environmental science/materials science

graphene oxide, heterotrophic nitrification–aerobic denitrification, nitrogen removal, monod model, low temperature

**Author for correspondence:**
Yuewei Yang
e-mail: yangyuewei@163.com

†Co-first author.

This article has been edited by the Royal Society of Chemistry, including the commissioning, peer review process and editorial aspects up to the point of acceptance.

# Denitrification performance of *Pseudomonas fluorescens* Z03 immobilized by graphene oxide-modified polyvinyl-alcohol and sodium alginate gel beads at low temperature

Meizhen Tang, Jie Jiang†, Qilin Lv, Bin Yang, Mingna Zheng, Xin Gao, Jindi Han, Yingjie Zhang and Yuewei Yang

Department of Environmental Science, School of Life Sciences, Qufu Normal University, Qufu 273165, People's Republic of China

JJ, 0000-0002-1095-3839

Improving the effect of microbial denitrification under low-temperature conditions has been a popular focus of research in recent years. In this study, graphene oxide (GO)-modified polyvinyl-alcohol (PVA) and sodium alginate (SA) (GO/PVA–SA) gel beads were used as a heterotrophic nitrification–aerobic denitrification (HN–AD) bacteria (*Pseudomonas fluorescens* Z03) carrier to enhance nitrogen removal efficiency levels at low temperatures (6–8°C). The removal efficiency of $NH_4^+$-N and $NO_3^-$-N and the variations in concentrations of extracellular polymeric substances (EPS) under different GO doses (0.03–0.15 g $l^{-1}$) were studied. The results indicated that the addition of GO can improve the efficiency of nitrogen removal, and the highest removal efficiency level and highest carbohydrate, protein, and total EPS content levels (50.28 mg, 132.78 mg and 183.06 mg (g GO/PVA–SA gel)$^{-1}$, respectively) were obtained with 0.15 g $l^{-1}$ GO. The simplified Monod model accurately predicted the nitrogen removal efficiency level. These findings suggested that the application of GO serves as an effective means to enhance nitrogen removal by stimulating the activity of HN–AD bacteria.

# 1. Introduction

The heterotrophic nitrification–aerobic denitrification (HN–AD) process is widely considered one of the most ground-breaking technological innovations for removing nitrogenous pollutants from sewage. This removal process involves heterotrophic microorganisms oxidizing ammonia nitrogen into nitrate nitrogen and nitrous nitrogen while nitrous nitrogen is reduced to nitrogen by anti-nitrification in the oxygenated environment. As a new, economical and efficient treatment method, a series of studies have been carried out on the characteristics of HN–AD since the discovery of a group of bacteria with HN–AD capabilities. So far, it has been proven that a variety of bacterial species exhibit depredating capabilities, including *Bacillus methylotrophicus* strain L7 [1], *Klebsiella pneumoniae* CF-S9 [2], *Vibrio diabolicus* SF16 [3], *Pseudomonas fluorescens* wsw-1001 [4], *Cupriavidus* sp. S1 [5], *Ochrobactrum* sp. HXN-1 and *Aquamicrobium* sp. HXN-2 [6]. With significant nitrogen removal capabilities, these functional bacterial strains also have a higher growth rate than autotrophs and can resist high organic loads [7,8]. Although the HN–AD process offers many potential advantages, some researchers have also indicated that the survival, growth and reproductive characteristics of microorganisms could be inhibited at low temperatures, which might consequently create difficulties for highly nitrogenous wastewaters in the biodegradation stage [9].

Bacteria immobilization remains an ideal form of wastewater treatment technology. Under numerous conditions, immobilized microbial cells achieve an improved biodegradation rate through a higher cell loading, and this bioprocess can be controlled more easily. It cannot only significantly increase the growth activity of microorganisms but also improve its tolerance of toxic or harmful substances produced during sewage treatment [10–12].

As a novel carbon material with excellent properties, graphene oxide (GO) sheets are mono-layers of carbon atoms with a carboxyl group, an epoxy group and a hydroxyl group on their bases and edges. Their extraordinary characteristics (e.g. low production costs, large specific surface area, high colloidal performance and low cytotoxicity [13,14]) have attracted considerable attention in the field of microbiology. A number of reports show that GO is a superior biocompatible material that can provide ideal growth environments with limited or no cytotoxicity to certain species of bacteria. For instance, when GO is placed on a filter it causes bacteria to grow faster, with bacteria growing 2 times and 3 times faster on filters coated with 25 and 75 μg than on filters without GO [15]. Therefore, GO has a large specific surface area, exhibits higher levels of hydrophilicity and reactive activity and could be considered to be a suitable carrier material for bacteria immobilization.

As noted above, research on HN–AD bacteria, as microorganisms with unique denitrification characteristics, has mainly focused on their characteristics, while less research has attempted to improve their denitrification performance, especially at low temperatures. This study is mainly focused on the HN–AD bacteria *Pseudomonas fluorescens* Z03 immobilized by GO-modified polyvinyl-alcohol (PVA) and sodium alginate (SA) (GO/PVA–SA) gel beads and investigates the positive effects of GO on *Pseudomonas fluorescens* Z03 in improving its nitrogen removal performance in low-temperature environments. In addition, the optimal dose of GO for PVA–SA gel beads for nitrogen removal and the specific effects of GO on *Pseudomonas fluorescens* Z03 are studied and discussed.

# 2. Material and methods

## 2.1. Graphene oxide synthesis

The synthetic method of modified GO refers to the bacteria-friendly method used in a previous work by Zhou *et al.* [12].

## 2.2. Microbial carrier synthesis and immobilization

*Pseudomonas fluorescens* Z03 was embedded during wastewater treatment with seven different types of microbial carriers, including PVA–SA and GO/PVA–SA at different concentrations. PVA and SA carrier solutions were prepared by dissolving 100.0 g of PVA in 500.0 ml of deionized water and 10.0 g of SA in 100.0 ml deionized water and stirring them in a 90°C water bath for 1 h and finally mixing the gel. A volume of graphene oxide stock solution ($1.5 \, \mathrm{g \, l^{-1}}$) was added to the 600 ml of PVA–SA carrier solution to prepare the GO/PVA–SA carrier solution with GO concentrations of 0.03, 0.05, 0.1, 0.15, 0.3 and $0.5 \, \mathrm{mg \, l^{-1}}$. Ultrasonic conditions were applied for 30 min (the ultrasonic

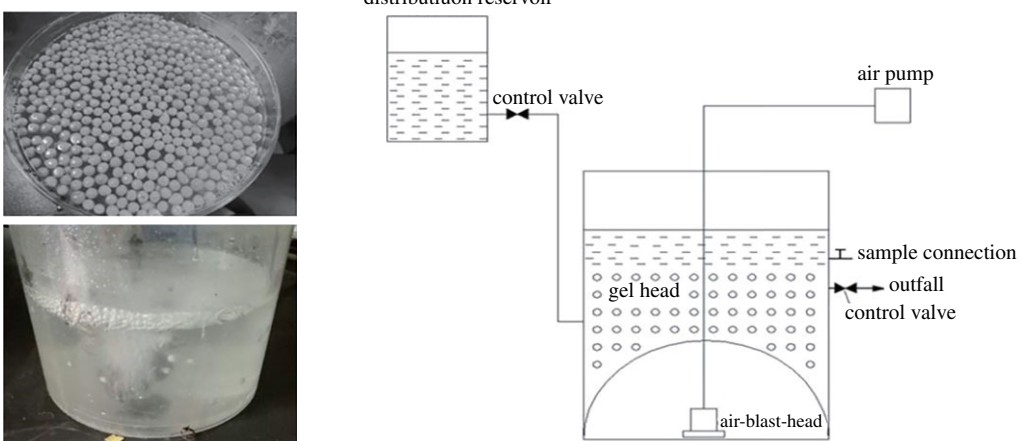

**Figure 1.** The gel beads used and a schematic diagram of the continuously stirred tank reactor.

frequency was set to 40 kHz), and the solution was cooled to 30°C. Then, 200 ml of enriched *Pseudomonas fluorescens* Z03 was mixed with the material solution and a volume of deionized water was added to obtain a total material solution volume of l litre that was stirred for 1 h. The mixture was added to a saturated boric acid solution of 2% of CaCl₂ with a particle size of approximately 3 mm through the granulator and was then cross-linked on the magnetic agitator for 30 min. Cross-linking embedding particles were then removed after 2–3 rounds of rinsing with physiological saline, stored in deionized water and preserved in a 4°C refrigerator.

## 2.3. Reactor operation, bacteria cultivation and wastewater treatment

A continuously stirred tank reactor with a total working volume of 1 l, a 10% (g (gel bead) ml$^{-1}$ (wastewater volume)) packing ratio of gel beads and synthetic wastewater was used (figure 1).

*Pseudomonas fluorescens* Z03 was isolated from wastewater biosludge provided by the Shandong Public Utility Group Qufu Water Co., Ltd. (116.970921E, 35.583021N).

The nitrogen wastewater was prepared by adding 0.228 g of glucose, 0.382 g of NH₄Cl, 0.01 g of MgSO₄·7H₂O, 0.304 g of NaNO₃, 0.0877 g of KH₂PO₄, 0.0076 g of CaCl₂·7H₂O and 0.0504 g of Na₂CO₃ in 1.0 l of deionized water. Ten grams of PVA–SA or GO/PVA–SA gel beads fixed with *Pseudomonas fluorescens* Z03 were separately added to 1.0 L of wastewater. The experimental temperature was set to 6–8°C and DO 2–4 mg l$^{-1}$. Water and GO/PVA–SA materials were sampled at regular intervals for analysis.

## 2.4. Analysis

Specific surface area and aperture were analysed with Gemini VII 2390p. Scanning electron microscope (SEM) images were obtained with JSM-6700F at 8.0 kV (JEOL, UK). Infrared (IR) spectroscopy involved employing the KBr tablet method and using 633 nm lasers to obtain Raman spectra through an inVia Raman microscope (RENISHAW, UK). Concentrations of ammonia, nitrite, nitrate, total nitrogen and COD were measured according to standard methods (APHA, 1998). Extracellular polymeric substances (EPS) of GO/PVA–SA gel samples were extracted by cationic exchange. Using glucose as a standard, the carbohydrate content of EPS was measured using the anthracene method. Using bovine serum albumin as a standard, the protein content of EPS was measured using the Lowry method [16].

# 3. Monod kinetic model

A simplified Monod kinetic model was used to simulate the denitrification efficiency of immobilized *Pseudomonas fluorescens* Z03 in a continuous stirred tank reactor [17]. It is assumed that the degradation of nitrogen is subordinate to Monod kinetics and that the reactor is a continuous stirred tank reactor (CSTR). Therefore, the Monod equation can be expressed as follows:

$$K = \frac{q(C_{in} - C_{out})(C_{half} + C_{out})}{C_{out}}, \tag{3.1}$$

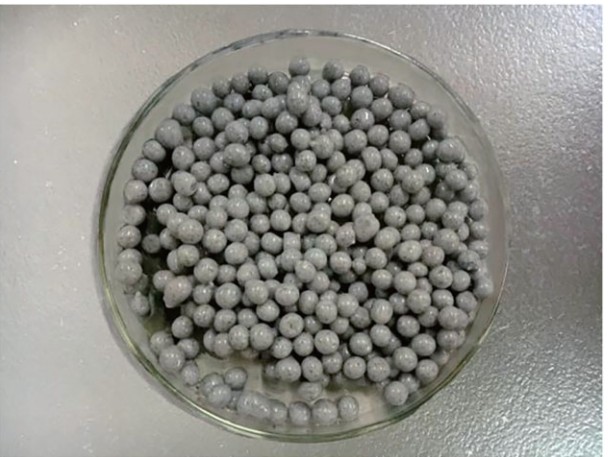

**Figure 2.** The modified gel beads.

**Table 1.** Apparent characteristics of the modified gel beads.

| characteristics of the modified gel beads | results |
|---|---|
| appearance | the modified gel beads are light black and approximately 3 mm in diameter; the modified gel beads are compact on the outside and porous on the inside with uniformly dispersed visible black and fine GO particles |
| density | the density of the modified gel beads was roughly $1.061 \times 10^3$ kg m$^{-3}$, which is close to the density of water, rendering them easy to fluidize |
| elasticity | particle gelation is high and elastic |
| mechanical strength | after considerable shock agitation, the modified gel beads did not break, indicating that their mechanical strength was high |
| conglutination | under the action of the magnetic agitator, spherical particles rapidly formed after the mixture was dripped into the cross-linking agent, there were no bonds between particles and the particles were uniform in shape and size |

where $K$ is the constant removal rate of the largest area (g · (m$^2$ d)$^{-1}$), q is the hydraulic load rate (m d$^{-1}$), $C_{in}$ is the nitrogen concentration of inflow (mg l$^{-1}$), $C_{half}$ is the limiting factor's half-saturation constant and $C_{out}$ is the nitrogen concentration of outflow (mg l$^{-1}$).

# 4. Results and discussion

## 4.1. Apparent properties of the modified gel beads

The modified gel beads under optimized preparation conditions are shown in figure 2, and their apparent characteristics are shown in table 1.

The BET method for nitrogen adsorption was used to determine the specific surface area and pore size of embedded particles before and after modification. The results are shown in table 2.

Table 2 shows that the specific surface area and total pore volume of the modified gel beads were substantially higher than those of the unmodified gel beads, increasing from 10.7284 m$^2$ g$^{-1}$ and 0.056725 cm$^3$ g$^{-1}$ to 13.5392 m$^2$ g$^{-1}$ and 0.072196 cm$^3$ g$^{-1}$, respectively.

In terms of pore size, the average pore size of the gel beads decreased from 11.03015 nm before modification to 7.4832 nm after modification. Studies have shown that the existence of holes and gaps can protect microorganisms from the effects of hydraulic disturbance and scour and that a decrease in pore size has less of an impact on hydraulic impact such that the strength of gel beads is enhanced.

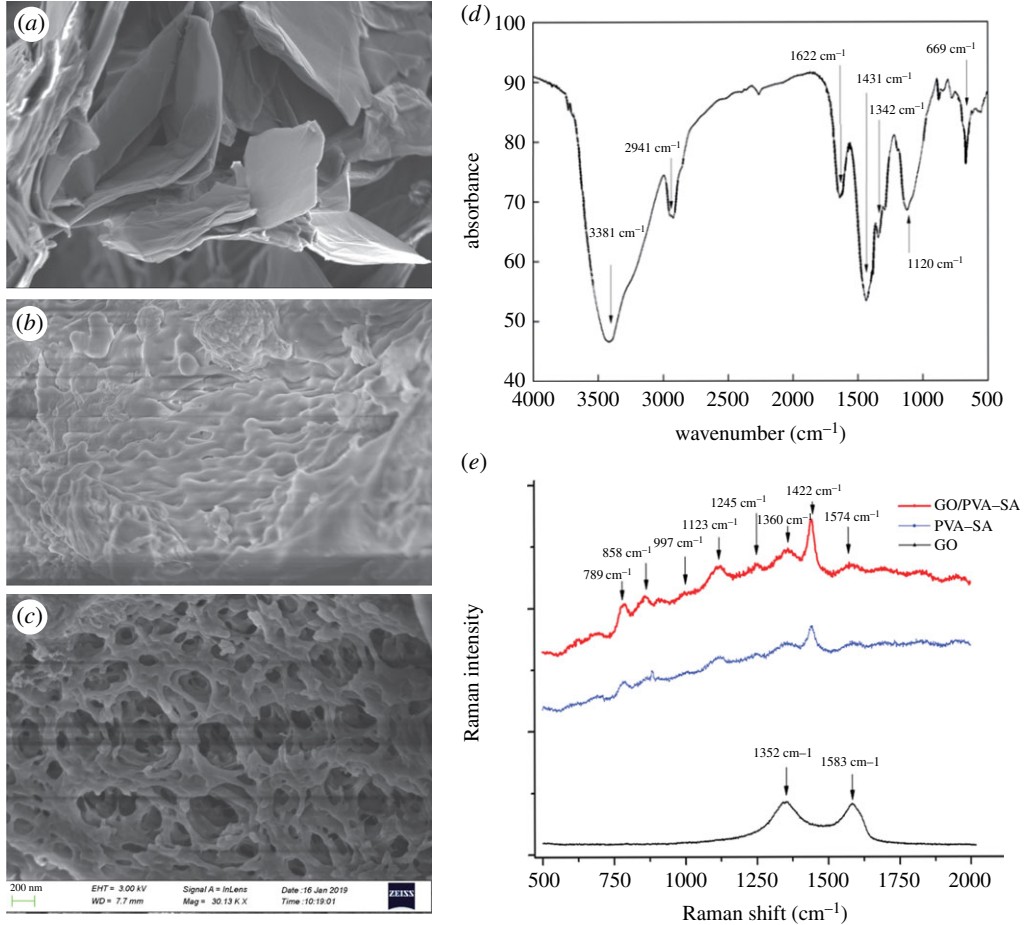

**Figure 3.** SEM images of GO (*a*), PVA–SA (*b*) and GO/PVA–SA (*c*) carriers. The infrared spectrogram (*d*) and Raman spectrum (*e*) of GO/PVA–SA materials.

**Table 2.** Pore structure parameters of the gel beads before and after modification.

| samples | specific surface area (m² g⁻¹) | total pore volume (cm³ g⁻¹) | average pore diameter (nm) | pore size range (nm) |
|---|---|---|---|---|
| PVA–SA | 10.7284 | 0.056725 | 11.3015 | 0.8–150.0 |
| GO-PVA–SA (0.15 g L⁻¹ GO) | 13.5392 | 0.072196 | 7.4832 | 0.8–150.0 |

On the other hand, *Pseudomonas fluorescens* Z03 bacteria used in this experiment were (0.6–0.8) µm × (1.1–1.4) µm in size, and their molecular and atomic diameters ranged 0.144–0.38 nm [18]. However, the modified gel beads were approximately 7.5 nm in diameter, which is smaller than the size of the bacteria but larger than the molecular diameter. Therefore, modified gel beads facilitated the efficient mass transfer of internal and external pollutants and metabolites, prevented the loss of immobilized microorganisms and promoted the growth and tolerance of microorganisms in the adverse environment.

## 4.2. Characterization of graphene oxide-modified polyvinyl-alcohol and sodium alginate material

SEM images of GO, PVA–SA and GO/PVA–SA are shown in figure 3*a–c*. The original GO did not have a pronounced reticular formation and was only observed in PVA–SA and improved in the GO/PVA–SA material. This result echoes those of the previous studies [19,20] showing PVA–SA gel beads to have a stable spherical shell with a honeycomb internal structure but with a compact surface structure. This structure is not suitable for bacterial attachment and proliferation and is also not conducive to mass transfer. By contrast, GO/PVA–SA gel beads have no obvious spherical shell with a remarkable

layered and very loose chemical structure formed by GO that can support bacterial immobilization, growth and faster mass transfer [21]. Therefore, it is recommended that GO/PVA–SA gel beads are designed with an advantageous microstructure with a high surface area so that microbes can be immobilized and facilitate faster mass transfer, in turn helping bacteria maintain high levels of biodegradability efficiency by tolerating low temperatures.

As is shown in figure 3*d*, the infrared spectrum of GO presents a strong absorption peak of 3381 cm$^{-1}$, which refers to the tensile vibration of -OH. The high hydrophilicity of PVA produces stretching vibrations among intermolecular and intramolecular hydrogen bonds [22]. Absorption peaks at 2941 cm$^{-1}$, 1622 cm$^{-1}$, 1431 cm$^{-1}$, 1342 cm$^{-1}$ and 1120 cm$^{-1}$ represent the vibrations of C–H, –COO, –CH$_2$ and C–O, consistent with data from past studies [22,23]. Similar results are shown by the Raman spectrum presented in figure 3*e*. PVA–SA has strong peaks of $\upsilon$(C–C) at 789 cm$^{-1}$, $\upsilon$(O–O) at 858 cm$^{-1}$, $\upsilon$(C–O–C) at 997 cm$^{-1}$ and 1123 cm$^{-1}$, and $\delta$(CH$_3$) at 1422 cm$^{-1}$. The G-band of the GO/PVA–SA material was set at 1574 cm$^{-1}$, and the D-band was set at 1360 cm$^{-1}$. Further evidence indicates the occurrence of close interactions between the hydrophilic functional group of the GO material and the PVA–SA matrix [24,25], whereby this structure supported bacterial immobilization for bio-augmentation.

## 4.3. Effect of different doses of graphene oxide in polyvinyl-alcohol and sodium alginate carriers on the nitrogen removal effect in one period

To study the characteristics of GO on the modified PVA–SA as a *Pseudomonas fluorescens* Z03 carrier, a single-period synthetic water treatment test was conducted to investigate its nitrogen removal effect in seven gel carriers. The experimental results are illustrated in figure 4*a–c* and tables 3 and 4.

In the control experiments without GO gel, NO$_3^-$-N and NH$_4^+$-N concentrations reduced from 100 mg l$^{-1}$ to 26.37 mg l$^{-1}$ and 36.85 mg l$^{-1}$, respectively, after 72 h of culturing (figure 4*a,b*). At the same time, the concentration of NO$_2^-$-N increased from 0 to 62.91 mg l$^{-1}$ over the first 36 h and then dropped to 0.013 mg l$^{-1}$ after 72 h (figure 4*c*). The fluctuation of concentrations of three different nitrogen forms proves that heterotrophic nitrification and aerobic denitrification were successfully executed in the experiments.

As is shown in tables 3 and 4, the concentrations of NH$_4^+$-N and NO$_3^-$-N show a significant correlation with GO doses in the GO/PVA–SA gel. Compared to that of PVA–SA carriers, the nitrogen removal efficiency of GO/PVA–SA gel (0.03–0.15 g l$^{-1}$ GO) material was found to be significantly higher. Specifically, when the GO dose in the GO/PVA–SA gel reached 0.15 g l$^{-1}$ GO, NH$_4^+$-N and NO$_3^-$-N concentrations, respectively, dropped to 3.62 and 0.91 mg l$^{-1}$ after 72 h of culturing, proving it to be the best carrier in these batch tests. Under the same conditions, when the GO dose was 0.3–0.5 g l$^{-1}$, the decline of NH$_4^+$-N and NO$_3^-$-N concentrations occurred rather slowly, while their removal rates reached 24.83% and 23.37%, respectively. This occurred because the GO/ PVA–SA carrier has better biological enhancement properties because GO has a highly specific surface area and functional groups support and stimulate bacterial growth. Although GO may have antibacterial effects [26], this problem could be resolved by surface modification, which could also promote the growth of certain strains. However, when the GO dose exceeded 0.3 g l$^{-1}$, this surface modification was no longer useful, and bacterial growth should have been suppressed, causing the nitrogen removal efficiency level to decline. In this work, the manufacturing process of the GO/PVA– SA carrier is considered to reduce the surface modification of GO to bacterial toxicity. The reaction between the GO carboxyl group and PVA–SA hydroxyl group forms a condensed layer, which provides space for bacterial adhesion and growth.

As is shown in figure 4, the amount of GO added has no effect on the intermediates of heterotrophic nitrification. Almost no accumulation of nitrite, nitrate or other intermediates occurs throughout the whole heterotrophic nitrification process, which is consistent with results found for *Pseudomonas putida* ZN1 [27] and *Klebsiella* sp. TN-10 [28].

The few accumulations of nitrates and nitrites observed signify that ammonia nitrogen might be converted directly into nitrogen gas, and N$_2$ production was measured by gas chromatography after 72 h of culturing, further verifying that strain Z03 can directly transform ammonium into nitrogen gas. Meanwhile, when ammonia nitrogen was the sole nitrogen source, 16.76–60.33% of the initial NH$_4^+$-N was transformed into N$_2$, 10.66–36.18% was converted into intracellular nitrogen and 3.49– 72.58% of ammonia nitrogen was left. This result shows that (i) strain Z03 can transform ammonium into N$_2$ directly, verifying its strong aerobic denitrification abilities, and (ii) the appropriate amount of

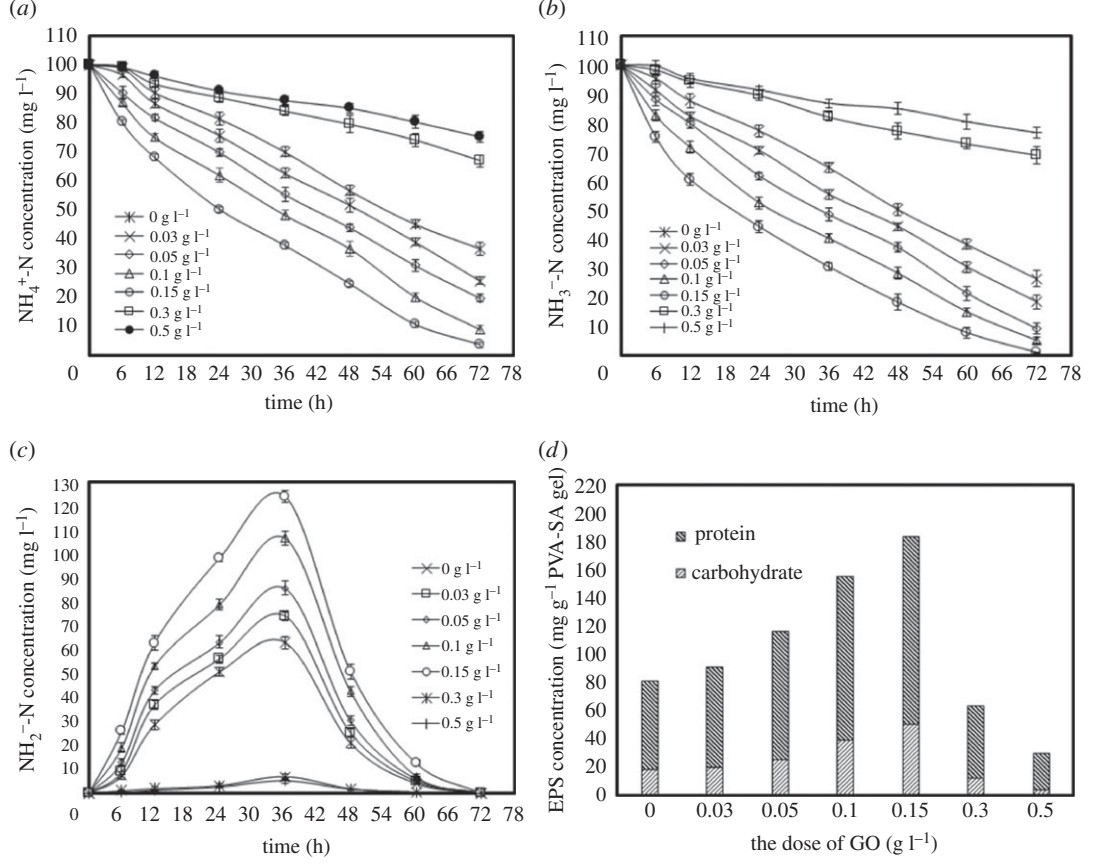

**Figure 4.** Concentrations of $NH_4^+$-N (*a*), $NO_2^-$-N (*b*), $NO_3^-$-N (*c*) and EPS (*d*) under the influence of various GO doses.

**Table 3.** Correlations between GO doses in GO/PVA–SA gel and $NH_4^+$-N concentrations at 72 h.

| | | GO dose in GO/PVA–SA gel | $NH_4^+$-N concentration |
|---|---|---|---|
| GO dose in GO/PVA–SA gel | Pearson Correlation | 1 | 0.757[a] |
| | Sig. (two-tailed) | | 0.049 |
| | N | 7 | 7 |
| $NH_4^+$-N concentration | Pearson Correlation | 0.757[a] | 1 |
| | Sig. (two-tailed) | 0.049 | |
| | N | 7 | 7 |

[a]Denotes that the correlation is significant at the 0.05 level (two-tailed).

**Table 4.** Correlations between GO doses in GO/PVA–SA gel and $NO_3^-$-N concentrations at 72 h.

| | | GO dose in GO/PVA–SA gel | $NO_3^-$-N concentration |
|---|---|---|---|
| GO dose in GO/PVA–SA gel | Pearson Correlation | 1 | 0.825 |
| | Sig. (two-tailed) | | 0.022 |
| | N | 7 | 7 |
| $NO_3^-$-N concentration | Pearson Correlation | 0.825[a] | 1 |
| | Sig. (two-tailed) | 0.022 | |
| | N | 7 | 7 |

[a]Denotes that the correlation is significant at the 0.05 level (two-tailed).

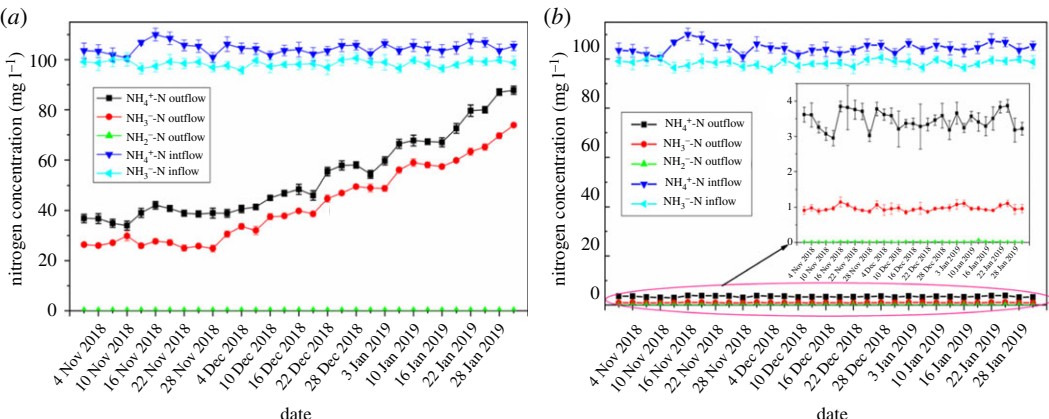

**Figure 5.** The long-term treatment effect of optimally modified gel beads fixed with *Pseudomonas fluorescens* Z03 on nitrogenous wastewater: (*a*) non-GO gel beads and (*b*) GO-modified gel beads (0.15 g l⁻¹ GO).

GO cannot only provide microorganisms with space to grow but can also stimulate the growth of microorganisms, thus improving its HN–AD ability.

## 4.4. Effects of graphene oxide on extracellular polymeric substances

Various microbial aggregates, such as biofilms, flocculants and sludge, are held together through EPS, and together, they play an important role in understanding the functions of microbial aggregation [29,30]. To further determine the effect of GO on the activity of *Pseudomonas fluorescens* Z03, EPS concentration variations at different GO doses were studied. As shown in figure 4*d*, the concentration of EPS increased within the range of 0.03 to 0.15 g l⁻¹ GO. In addition, under the combined action of 0.15 g l⁻¹ GO and *Pseudomonas fluorescens* Z03, the increase in carbohydrate, protein and total EPS production reached a maximum (50.3, 132.9 and 183.2 mg·g of GO/PVA–SA gel⁻¹, respectively). However, when the GO dose was increased to 0.3–0.5 g l⁻¹, yields of carbohydrate, protein and total EPS decreased. These results indicate that GO has a significant effect on the production of EPS and plays an important role in the study of microbial bioactivity [31].

As is indicated in previous reports, due to the special characteristics of the polymers' rich matrix structures, EPS include numerous oxygen-containing functional groups (e.g. –COOH, –NH, –OH and –CO–). EPS also shape the formation of multilevel porous structures, in which smaller microcolonies can form macroflocs through the cohesion of microorganisms [31–33]. In this experiment, the volumes of EPS increased when the GO dose was in the range of 0.03–0.15 g l⁻¹. It could be roughly inferred that GO has a positive effect on *Pseudomonas fluorescens* Z03 in producing EPS, which leads to the formation of more macroscopic flocculants in GO/PVA–SA gel beads and enhances the activity of *Pseudomonas fluorescens* Z03. However, when GO doses increased to 0.3–0.5 g l⁻¹, EPS production greatly decreased, which can be ascribed to structural damage to and characteristic changes in bacteria [34–36].

## 4.5. Long-term impacts of optimal modified gel beads on the nitrogen removal effect

A long-term synthetic water treatment test was conducted to evaluate the impacts of the optimized modified gel beads on the nitrogen removal effect. The results are recorded and illustrated in figure 5.

As shown in figure 5, the best GO-modified gel beads (0.15 g l⁻¹) fixed with *Pseudomonas fluorescens* Z03 for three consecutive months (water temperatures of 6–8°C and DO of 2–4 mg l⁻¹) have a stable denitrification effect. The removal efficiencies of $NH_4^+$-N and $NO_3^-$-N were measured as 96.38–97.24% and 98.82–99.12%, respectively, and the cumulant of $NO_2^-$-N was very low at less than 0.1 mg l⁻¹. By contrast, the denitrification effect of the control (non-GO gel beads) consistently decreased overtime. With continuous processing for 31 days, the removal efficiency of $NH_4^+$-N and $NO_3^-$-N reached only 16.69% and 25.26%, respectively, as the non-GO gel beads became swollen and damaged by long-term effects of aeration agitation, causing *Pseudomonas fluorescens* Z03 to be lost with the outflow. Meanwhile, the removal efficiency levels of $NH_4^+$-N and $NO_3^+$-N in optimal GO-modified gel beads (0.15 g l⁻¹ GO) were 5.81 and 3.92 times those of the control (non-GO gel bead). Furthermore, the

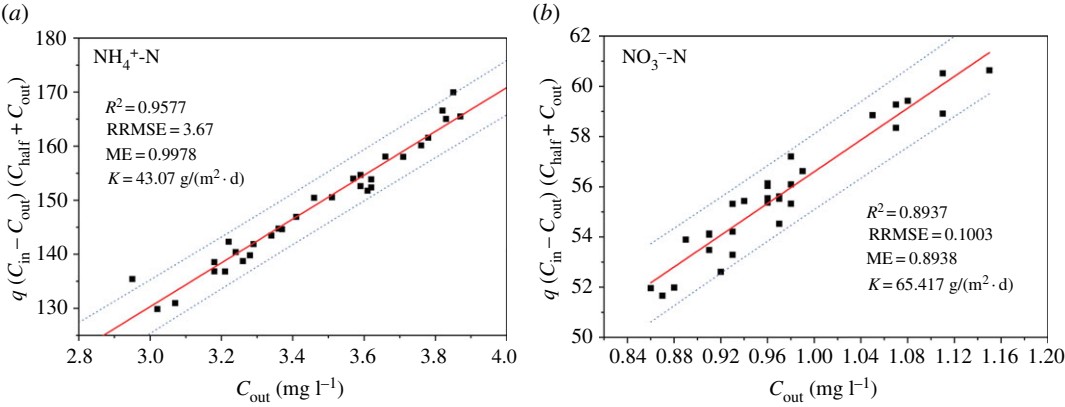

**Figure 6.** Monod kinetics regression of inlet and outlet (*a*) $NH_4^+$-N and (*b*) $NO_3^-$-N values in the continuously stirred tank reactor.

average concentrations of $NH_4^+$-N, $NO_3^-$-N and $NO_2^-$-N in the effluent reached 3.46, 0.97 and 0.019 mg l$^{-1}$, respectively, falling below grade A status according to the Chinese national pollutant discharge standard for municipal wastewater treatment plants (GB18918-2002).

The highly stable denitrification effect may have occurred for the following reasons: (i) GO rendered the gel bead surfaces rougher and pore structures more abundant and increased specific surface areas and the number of micropores, providing space for *Pseudomonas fluorescens* Z03 to inhabit and reproduce, preventing it from being removed through the outflow of water and improving its tolerance of the environment (e.g. pH levels, temperatures, toxic substances, etc.). Thus, the stability of nitrogen removal in low-temperature wastewater was improved. (ii) Low concentrations of GO had a positive effect on the growth of *Pseudomonas fluorescens* Z03, corroborating previous studies [15,24,37].

## 4.6. Simulation and validation of the model

Relative to the pseudo-first-order kinetic model, the Monod kinetic model conforms better to the actual conditions of microbial treatment, and thus it is more applicable to the degradation process of pollutants, in which microorganisms play a leading role [38]. When the abovementioned Monod model was used to simulate the nitrogen removal efficiency of optimal modified gel beads fixed with *Pseudomonas fluorescens* Z03 in the continuously stirred tank reactor, the $C_{half}$ values of this model for $NH_4^+$-N and $NO_3^-$-N were 1.0 mg l$^{-1}$ [39] and 0.76 mg l$^{-1}$, respectively [40]. Therefore, the following three parameters were introduced to estimate the advantages of this model:

$$\text{Coefficient determination:} \quad R^2 = \frac{\left[\sum_{i=1}^{N}(X_i - \overline{X})(Y_i - \overline{Y})\right]^2}{\sum_{i=1}^{N}(X_i - \overline{X})^2 \sum_{i=1}^{N}(Y_i - \overline{Y})^2} \quad \text{(value range: 0–1),} \tag{4.1}$$

$$\text{Relative root-mean-square error:} \quad \text{RRMES} = \frac{\sqrt{(1/N)\sum_{i=1}^{N}(Y_i - \hat{Y})^2}}{\hat{Y}} \quad \text{(value range: 0–∞)} \tag{4.2}$$

$$\text{and} \qquad \text{Model efficiency:} \quad \text{ME} = \frac{1 - \sum_{i=1}^{N}(Y_i - \hat{Y})^2}{\sum_{i=1}^{N}(Y_i - \overline{Y})^2} \quad \text{(value range: −∞–0),} \tag{4.3}$$

where $X_i$ and $Y_i$ were the nitrogen concentrations in the inflow and outflow at different times, respectively; $\overline{X}$ and $\overline{Y}$ were the mean nitrogen concentrations in the inflow and outflow, respectively, and $\hat{Y}$ was the predictive value.

As shown in figure 6, the simplified Monod model reveals the relationship between $f(C_{in}, C_{out}, q)$ and $C_{out}$. The maximum removal rate for nitrogen (K, g m$^{-2}$ d$^{-1}$) equals the slope of the regression line, and the deviation between the measured and predicted values of nitrogen is indicated by statistical parameters ($R^2$, RRMSE and ME). The dotted line represents the 95% confidence band and reflects the real regression line.

Figure 6 shows that statistical parameter $R^2$ for $NO_3^-$-N and $NH_4^+$-N was valued at 0.8937 and 0.9577, respectively, and RRMSE and ME values fell within the expected range. The study results reveal a good fit between the predicted and experimental values of the nitrogen removal rate when using optimized modified gel beads fixed with *Pseudomonas fluorescens* Z03 in the continuously stirred

tank reactor. Degradation rates were identified when pollutants were used as the sole energy or carbon source, and the relationship between rates and biomass generated by every unit of energy or carbon source consumed was also investigated [41]. Moreover, when the contaminants were at a controlled concentration, the model was also able to describe the degradation rate with different microorganism doses. Therefore, this simplified Monod model can accurately and efficiently predict the nitrogen removal rate.

# 5. Conclusion

In this study, GO/PVA–SA gel beads were used as *Pseudomonas fluorescens* Z03 carriers to enhance nitrogen removal efficiency levels at low temperatures (6–8°C). GO was dispersed into PVA–SA gel beads and stimulated the bioactivity of *Pseudomonas fluorescens* Z03 immobilized by PVA–SA. When GO was added at 0.15 g l$^{-1}$, the removal efficiency levels of $NH_4^+$-N and $NO_3^-$-N were 96.38–97.24% and 98.82–99.12%, respectively, and the cumulant of $NO_2^-$-N was very low at below 0.1 mg l$^{-1}$. The highest volumes of carbohydrate, protein and total EPS were also obtained under this value. In addition, an appropriate GO dose can be effective in stimulating an increase in EPS. The simplified Monod model accurately predicted the nitrogen removal efficiency level with optimally modified gel beads fixed with *Pseudomonas fluorescens* Z03 in a continuously stirred tank reactor. Therefore, the application of GO constitutes an effective means to enhance nitrogen removal by stimulating the activity of HN–AD bacteria.

Data accessibility. The batch experimental data used are available from the Dryad Digital Repository: https://dx.doi.org/10.5061/dryad.fn2z34tps [42].

Authors' contributions. B.Y., M.Z. and X.G. performed the experiments and participated in data analysis; M.T. and Q.L. contributed to the design of the study and drafted the manuscript; J.H. and Y.Z. collected field data; M.T. and Y.Y. conceived of and designed the study; M.T. and J.J. carried out the statistical analyses, coordinated study tasks and helped draft the manuscript. All authors have given final approval to publish the article.

Competing interests. We declare we have no competing interests.

Funding. This work was supported by the National Natural Science Foundation of China (grant nos. 31700433 and 31672314).

Acknowledgements. The authors gratefully acknowledge financial support provided by the National Natural Science Foundation of China (no. 31700433; no. 31672314).

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
