## [Reviewer comments · Royal Society Open Science]

Review History

RSOS-191542.R0 (Original submission)

Review form: Reviewer 1

Is the manuscript scientifically sound in its present form?

Yes

Are the interpretations and conclusions justified by the results?

Yes

Is the language acceptable?

No

Do you have any ethical concerns with this paper?

No

Have you any concerns about statistical analyses in this paper?

No

Recommendation?

Major revision is needed (please make suggestions in comments)

Comments to the Author(s)

This work reported the denitrification performance of pseudomonas fluorescens Z03 immobilized by GO modified PVA-SA gel bead at low temperature. Based on the evaluation of its originality, significance of content, quality of the presentation, scientific soundness, and interest to readers, major revision is suggested before the article may be considered for acceptance. Specific suggestions are provided below.

1. Line 23-24, "under cold temperature": it is better to add specific temperature value here.
2. Figure 3d, the title of X-axis should be wavenumber, rather than wavelength.
3. Figure 4a-c, the titles of Y-axis maybe is wrong. Please check them. (maybe should be removal efficiency (%))
4. Please keep the unit form consistent. (always use g/L or g L-1)
5. The quality of all figures should be improved to ensure all charaters in the figure are readable. An appropriate increase of the font size is necessary.
6. The quality of abstract should be improved. A high-quality abstract should include the following elements: Background, Objectives, Methods, Results, and Conclusions, and be a meaningful and accurate representation of the article.
7. Substantial English (language and style) improvement is required.

Review form: Reviewer 2

Is the manuscript scientifically sound in its present form?

No

Are the interpretations and conclusions justified by the results?

No

Is the language acceptable?

No

Do you have any ethical concerns with this paper?

No

Have you any concerns about statistical analyses in this paper?

Yes

Recommendation?

Reject

Comments to the Author(s)

Review for RSOS-191542

The authors reported denitrification performance of pseudomonas fluorescens Z03 immobilized by a gel bead at low temperature. Though some of the modeling has some useful information, the overall novelty of the current work is not good. The information on the immobilization of the bacterium is very limited.

Statistical analysis should be conducted and reported (Table 2, for instance). Only showing individual data has no meaningful significance. Then why conduct experiments with different groups? No statistical analysis, no conclusion you can make.

References: should be checked carefully. Line 387: actually the names are first names rather than family names. Original full text articles should be read and double-checked.

Review form: Reviewer 3

Is the manuscript scientifically sound in its present form?

Yes

Are the interpretations and conclusions justified by the results?

No

Is the language acceptable?

No

Do you have any ethical concerns with this paper?

No

Have you any concerns about statistical analyses in this paper?

No

Recommendation?

Major revision is needed (please make suggestions in comments)

Comments to the Author(s)

Comments to RSOS-191542:

The removal efficiency 329 of NH_4^+-N and $\text{NO}_3^- - \text{N}$ over designed GO loaded PVA-SA beads has been investigated under various testing conditions. Surely speaking, the resultant samples were characterized using various techniques and the optimal conditions were established. I can recommend it to be considered for publication in the journal after major revision as suggested by following items.

1. The dispersion status of GO species in the beads should be confirmed.
2. Comparative study between those reported studies should be provided, to let the readers understand the interest and enhancement of this work.
3. English writing should be improved in the text.

Decision letter (RSOS-191542.R0)

06-Jan-2020

Dear Miss Jiang:

Title: Denitrification performance of pseudomonas fluorescens Z03 immobilized by GO modified PVA-SA gel bead at low temperature
Manuscript ID: RSOS-191542

The editor assigned to your manuscript has now received comments from reviewers. We would like you to revise your paper in accordance with the referee and Subject Editor suggestions which can be found below (not including confidential reports to the Editor). Please note this decision does not guarantee eventual acceptance.

Please submit your revised paper before 29-Jan-2020. Please note that the revision deadline will expire at 00.00am on this date. If we do not hear from you within this time then it will be assumed that the paper has been withdrawn. In exceptional circumstances, extensions may be possible if agreed with the Editorial Office in advance. We do not allow multiple rounds of revision so we urge you to make every effort to fully address all of the comments at this stage. If deemed necessary by the Editors, your manuscript will be sent back to one or more of the original reviewers for assessment. If the original reviewers are not available we may invite new reviewers.

RSC Associate Editor:
Comments to the Author:
According to the comments of the adjudicator, the decision was made.

RSC Subject Editor:
Comments to the Author:
(There are no comments.)

Reviewers' Comments to Author:

Reviewer: 1

Comments to the Author(s)

This work reported the denitrification performance of *pseudomonas fluorescens* Z03 immobilized by GO modified PVA-SA gel bead at low temperature. Based on the evaluation of its originality, significance of content, quality of the presentation, scientific soundness, and interest to readers, major revision is suggested before the article may be considered for acceptance. Specific suggestions are provided below.

1. Line 23-24, "under cold temperature": it is better to add specific temperature value here.
2. Figure 3d, the title of X-axis should be wavenumber, rather than wavelength.
3. Figure 4a-c, the titles of Y-axis maybe is wrong. Please check them. (maybe should be removal efficiency (%))
4. Please keep the unit form consistent. (always use g/L or g L⁻¹)
5. The quality of all figures should be improved to ensure all characters in the figure are readable. An appropriate increase of the font size is necessary.
6. The quality of abstract should be improved. A high-quality abstract should include the following elements: Background, Objectives, Methods, Results, and Conclusions, and be a meaningful and accurate representation of the article.
7. Substantial English (language and style) improvement is required.

Reviewer: 2

Comments to the Author(s)

Review for RSOS-191542

The authors reported denitrification performance of *pseudomonas fluorescens* Z03 immobilized by a gel bead at low temperature. Though some of the modeling has some useful information, the overall novelty of the current work is not good. The information on the immobilization of the bacterium is very limited.

Statistical analysis should be conducted and reported (Table 2, for instance). Only showing individual data has no meaningful significance. Then why conduct experiments with different groups? No statistical analysis, no conclusion you can make.

References: should be checked carefully. Line 387: actually the names are first names rather than family names. Original full text articles should be read and double-checked.

Reviewer: 3

Comments to the Author(s)

Comments to RSOS-191542:

The removal efficiency 329 of NH₄⁺-N and NO₃⁻-N over designed GO loaded PVA-SA beads has been investigated under various testing conditions. Surely speaking, the resultant samples were characterized using various techniques and the optimal conditions were established. I can recommend it to be considered for publication in the journal after major revision as suggested by following items.

1. The dispersion status of GO species in the beads should be confirmed.
2. Comparative study between those reported studies should be provided, to let the readers understand the interest and enhancement of this work.
3. English writing should be improved in the text.

Author's Response to Decision Letter for (RSOS-191542.R0)

See Appendix A.

Decision letter (RSOS-191542.R1)

04-Feb-2020

Dear Miss Jiang:

Title: Denitrification performance of *Pseudomonas fluorescens* Z03 immobilized by GO-modified PVA-SA gel beads at low temperature
Manuscript ID: RSOS-191542.R1

It is a pleasure to accept your manuscript in its current form for publication in Royal Society Open Science. The chemistry content of Royal Society Open Science is published in collaboration with the Royal Society of Chemistry.

RSC Associate Editor
Comments to the Author:
(There are no comments.)

Reviewer(s)' Comments to Author:

Appendix A

Dear Dr Laura Smith and reviewers,

Thank you for your letter and the reviewer's comments on our manuscript entitled "Denitrification performance of *Pseudomonas fluorescens* Z03 immobilized by GO modified PVA-SA gel bead at low temperature" (Manuscript ID: RSOS-191542). Those comments are very helpful for revising and improving our paper, as well as the important guiding significance to other research. We have studied the comments carefully and made corrections which we hope meet with approval. The main corrections are in the manuscript and the responds to the reviewers' comments are as follows.

Replies to the reviewers' comments:

Replies to Reviewer 1

1. Line 23-24, "under cold temperature": it is better to add specific temperature value here.

Response: As reviewer suggested that we have added specific temperature value (6-8°C) in revised manuscript.

2. Figure 3d, the title of X-axis should be wavenumber, rather than wavelength.

Response: We are very sorry for our negligence and we have made correction according to the Reviewer's comments.

3. Figure 4a-c, the titles of Y-axis maybe is wrong. Please check them. (maybe should be removal efficiency (%))

Response: We have checked Figure 4 and made corrections.

4. Please keep the unit form consistent. (always use g/L or g·L⁻¹)

Response: We have made correction according to the Reviewer's comments.

5. The quality of all figures should be improved to ensure all characters in the figure are readable. An appropriate increase of the font size is necessary.

Response: As reviewer suggested that we have adjusted our figures to ensure all characters in the figure are readable.

6. The quality of abstract should be improved. A high-quality abstract should include the following elements: Background, Objectives, Methods, Results, and Conclusions, and be a meaningful and accurate representation of the article.

Response: As reviewer suggested that we have rewritten the abstract.

7. Substantial English (language and style) improvement is required.

Response: Thank you for your comments. In order to improve the quality of the article, we used AJE's English editing service to edit the manuscript.

Special thanks to you for your good comments.

Replies to Reviewer 2

The authors reported denitrification performance of pseudomonas fluorescens Z03 immobilized by a gel bead at low temperature. Though some of the modeling has some useful information, the overall novelty of the current work is not good. The information on the immobilization of the bacterium is very limited.

Statistical analysis should be conducted and reported (Table 2, for instance). Only showing individual data has no meaningful significance. Then why conduct experiments with different groups? No statistical analysis, no conclusion you can make.

References: should be checked carefully. Line 387: actually, the names are first names rather than family names. Original full text articles should be read and double-checked.

Response:

Thank you for your comments.

As reviewer suggested that we have supplemented the research background on microbial immobilization in the second paragraph of the introduction, and added table 3 and table 4 to do the correlation analysis of the concentration of $\text{NH}_4^+\text{-N}$ and $\text{NO}_3^-\text{-N}$ and the GO dosage of GO/PVA-SA gel, and analysis results in line 223-225 of the revised manuscript.

We are very sorry for our negligence of the format of the reference. We have checked and revised the full text articles carefully according to the submission guidance.

Special thanks to you for your good comments.

Replies to Reviewer 3

1. The dispersion status of GO species in the beads should be confirmed.

Response: In our study, GO solution was added into the 600 mL PVA-SA carrier solution to prepare the GO/PVA-SA carrier solution. Ultrasonic 30 minutes to mix the solution before make the mixture into beads. Therefore, GO is uniformly dispersed in the beads, and this is also the reason why the GO modified gel beads show a uniform light black. As reviewer suggested that we have add this part in table 1.

2. Comparative study between those reported studies should be provided, to let the readers understand the interest and enhancement of this work.

Response: Thank you for your comments. We have rewritten the abstract section, emphasizing the focus of our research, and added the innovation of our research to the last paragraph of the introduction, highlighting the differences from previous related research. In addition, the differences between our research and other reported studies are also reflected in the results and discussion.

3. English writing should be improved in the text.

Response: Thank you for your comments. In order to improve the quality of the article, we used AJE's English editing service to edit the manuscript.

Special thanks to you for your good comments.

We tried our best to improve the manuscript and made some changes in the manuscript.

Many grammatical or typographical errors have been revised. These changes will not

influence the content and framework of the paper. We appreciate for Editors/Reviewers' warm work earnestly, and hope that the correction will meet with approval.

Once again, thank you very much for your comments and suggestions.

Yours sincerely,

Meizhen Tang, Jie Jiang, Qilin Lv, Bin Yang, Mingna Zheng, Xin Gao, Jindi Han, Yingjie Zhang, Yuewei yang